# Reproductive system, temperature, and genetic background effects in experimentally evolving populations of *Caenorhabditis elegans*

**Joanna K. Baran**[1], **Paulina Kosztyła**[1], **Weronika Antoł**[1,2], **Marta K. Labocha**[1], **Karolina Sychta**[2], **Szymon M. Drobniak**[1,3], **Zofia M. Prokop**[1] *

**1** Faculty of Biology, Institute of Environmental Sciences, Jagiellonian University in Krakow, Krakow, Poland,
**2** Institute of Systematics and Evolution of Animals, Polish Academy of Sciences, Krakow, Poland,
**3** Evolution and Ecology Research Centre, School of Biological, Earth and Environmental Sciences, University of New South Wales, Sydney, NSW, Australia

* z.m.prokop@gmail.com

## Abstract

Experimental evolution (EE) is a powerful research framework for gaining insights into many biological questions, including the evolution of reproductive systems. We designed a long-term and highly replicated EE project using the nematode *C. elegans*, with the main aim of investigating the impact of reproductive system on adaptation and diversification under environmental challenge. From the laboratory-adapted strain N2, we derived isogenic lines and introgressed the fog-2(q71) mutation, which changes the reproductive system from nearly exclusive selfing to obligatory outcrossing, independently into 3 of them. This way, we obtained 3 pairs of isogenic ancestral populations differing in reproductive system; from these, we derived replicate EE populations and let them evolve in either novel (increased temperature) or control conditions for over 100 generations. Subsequently, fitness of both EE and ancestral populations was assayed under the increased temperature conditions. Importantly, each population was assayed in 2–4 independent blocks, allowing us to gain insight into the reproducibility of fitness scores. We expected to find upward fitness divergence, compared to ancestors, in populations which had evolved in this treatment, particularly in the outcrossing ones due to the benefits of genetic shuffling. However, our data did not support these predictions. The first major finding was very strong effect of replicate block on populations' fitness scores. This indicates that despite standardization procedures, some important environmental effects were varying among blocks, and possibly compounded by epigenetic inheritance. Our second key finding was that patterns of EE populations' divergence from ancestors differed among the ancestral isolines, suggesting that research conclusions derived for any particular genetic background should never be generalized without sampling a wider set of backgrounds. Overall, our results support the calls to pay more attention to biological variability when designing studies and interpreting their results, and to avoid over-generalizations of outcomes obtained for specific genetic and/or environmental conditions.

**Data Availability Statement:** The data may be found here: https://github.com/szymekdr/230206_Celeg_ee (section "Data").

**Funding:** The current Funding Statement (National Science Centre, Poland, grants OPUS (2013/09/B/NZ8/03317) and Sonata Bis (UMO-2017/26/E/NZ8/00879) to Z.M.P.) is correct. The funders had no role in study design, data collection and analysis, decision to publish, or preparation of the manuscript.

**Competing interests:** The authors have declared that no competing interests exist.

## Introduction

*"Evolutionary biologists are still fascinated by-and struggling to understand-the dynamics of adaptation and diversification, especially for those traits that affect the reproductive success of individual organisms. How quickly do populations change in these traits, and are their rates of change constant or variable? How rapidly do populations diverge from one another in these traits, and are rates of adaptation and diversification tightly or loosely coupled? How repeatable is evolution (. . .)? How do the answers to these questions depend on the genetic system of an organism (. . .)?"* [1]

Experimental evolution (EE) has been recognised a powerful research framework for gaining insights into wide range of biological questions. The power of this approach lies in using replicated and controlled experiments to directly track, over generations, changes in fitness (or/and any other traits of interest), occurring in response to specified environmental conditions [2]. Over nearly 3 decades since the seminal paper by Lenski & Travisano [1], quoted above, was published, experimental evolution studies have contributed hugely to our current understanding of the issues outlined in that quote. Research on microbial organisms has been particularly fruitful because biological properties of microbes enable maintaining large experimental populations in many replicates, thousands of generations spanned in a relatively short time, and fitness assays in which derived lineages can be directly competed against their ancestors (e.g. [1]). However, multicellular organisms differ profoundly from microbes by much greater complexity (both phenotypic and genomic) and by typically exhibiting (some form of) sexual mode of reproduction. Not only do both these features–complexity and sexual reproduction–belong to long-standing puzzles of evolutionary biology (e.g. [3–5]); they are also expected to have considerable impact on evolutionary processes themselves (e.g. [5–7]). Consequently, it is clear that insights concerning the dynamics of adaptive evolution, derived from microbial studies, cannot be easily extrapolated on multicellular organisms. Meanwhile, experimental evolution studies on multicellular organisms are much more challenging in terms of number of generations that can be spanned as well as the size and number of populations that can be maintained. Also, in most multicellular organisms, fitness is extremely difficult (or even logistically impossible) to measure, and competing derived lineages against their common ancestor is impossible.

From that perspective, the nematode *Caenorhabditis elegans* has become an exceptionally attractive multicellular model for experimental evolution [8, 9] because it has short generation time (3–4 days in standard laboratory conditions), is easy to culture in large numbers, and can be frozen and stored at -80˚C for extended period of time, then defrosted and brought back to activity, enabling direct comparisons of derived lines with their ancestral populations. Moreover, its reproductive system can be genetically manipulated. Wild type *C. elegans* populations are composed predominantly of hermaphrodites, which can self-fertilize but cannot cross-fertilize each other, and (very rare, usually < 0,5%) males, which can fertilize hermaphrodites. Thus, *C. elegans* can reproduce via two distinct systems: selfing and outcrossing. Outcrossing involves the fusion of gametes coming from two separate individuals, resulting in the mixing of two different genomes. Selfing, on the other hand, while technically also representing sexual reproduction (since it involves meiosis and subsequent gamete fusion), is uniparental and does not entail the mixing of separate genomes. Thus, it bears a number of important similarities to asexuality [10, 11]. While selfing is a predominant reproductive mode in most natural and laboratory populations of *C. elegans*, a number of mating system-altering mutations are known ([12–15]; for review, see [16]), including several which induce obligate outcrossing by disrupting sperm production in hermaphrodites. Altering the model's reproductive system gives researchers unique opportunities for experimental investigations of its impact on

evolutionary trajectories and outcomes. In particular, it can be applied to questions concerning the evolutionary maintenance of sexual outcrossing which, as mentioned above, has been one of the long-standing puzzles in evolutionary biology.

Under outcrossing, a parent passes only 50%, instead of a 100%, of its genes onto each offspring, hence suffering from a two-fold transmission decrease [3]. Moreover, the majority of sexually reproducing animals produce males and hence–theoretically at least–should also suffer the two-fold cost in terms of population-level fitness (often called two-fold cost of sex or two-fold cost of males; [17]). Despite these costs, the bulk of animals use outcrossing as reproductive mode, which calls for explanation. Most evolutionary hypotheses regarding the pervasiveness of outcrossing evoke the role of genetic shuffling, which increases genetic variation and can break selection interference between beneficial and deleterious mutations (Hill-Robertson effect), facilitating the spread of the former and the purging of the latter (reviewed by [3]). These effects can lead to increase in fitness, especially in changing environments. Additionally, the positive impact of outcrossing on adaptation may also arise due to sexual selection acting on males. This could happen if males which are better adapted, i.e., carry alleles beneficial in the novel environment, achieve the highest reproductive success. Under such scenario, sexual selection would work in the same direction as other components of natural selection, increasing its power to produce adaptation (reviewed by [18]). Empirical evidence that outcrossing facilitates adaptation is growing (e.g. [11, 19–21]). However, not all studies show this effect (e.g. [22]).

Largely inspired by the long term evolution experiment of Richard Lenski's group, including the influential paper quoted above [1], we designed a long-term and highly replicated experimental evolution project using *C. elegans*. The primary aim of this study was investigating the impact of reproductive system (almost exclusively selfing wild type vs. obligatory outcrossing) on adaptation to stressful environmental change (elevated temperature). As a starting population, we used laboratory adapted *C. elegans* strain N2 [23] obtained from Caenorhabditis Genetics Center (CGC). From the N2 strain we derived isogenic lines and subsequently introgressed the *fog-2(q71)* mutation, which enforces obligatory outcrossing by blocking sperm production in hermaphrodites, into three of them (see Method section). This way we obtained three pairs of isogenic ancestral populations differing in reproductive system. From each of the 6 (3 isolines × 2 reproductive systems) isogenic ancestral populations, we derived replicate lines and let them evolve under either increased or control temperature–thus creating a 2 by 2 by 3 design of two reproductive systems × two environmental treatments × three genetic backgrounds.

The initial shortage of genetic variation was important for two reasons. The first pertains to the biology of our model system: low levels of standing genetic variation are generally characteristic of *C. elegans* populations, due to the species' primarily selfing mode of reproduction, which enables (nearly) clonal expansions of single genotypes and associated genome-wide selective sweeps [24]. Secondly, we wanted to minimize initial differences in genetic background between wild type vs. obligatorily outcrossing ancestral populations. If genetically diverse starting populations were used, such differences would inevitably arise during introgression of *fog-2(q71)* mutation. Starting experimental evolution from isogenic populations meant that adaptation could only emerge from new mutations. This limits adaptive potential considerably, compared to situations in which standing genetic variation is present. However, rapid evolution from new mutations is possible under strong selection, and has in fact been observed in multiple studies, including on *C. elegans*. For example, Denver et al. [25] tracked five independent populations derived from a low-fitness isogenic progenitor and evolving at population size of *ca.* 1000 individuals for 60 generations. All five populations showed rapid increase in fitness, which subsequent genomic analysis showed to be associated with very fast fixation of new mutations. Teotonio et al. [26] found signatures of adaptation after 100

generations of evolution in replicated populations starting both from high and low (inbred) genetic diversity. Azevedo et al. [27] selected two initially isogenic populations on body size, in opposite directions, for 48 generations, resulting in 35% increase and 8% decrease of body size in the up-selected and down-selected line, respectively. Altogether, *C. elegans* biology indicates that evolution from new genetic variants, rather than from standing genetic diversity, is likely the main avenue for adaptation in this species, whereas empirical evidence described by other researchers indicates that such adaptation can proceed rapidly. Indeed, in two previously published studies from the same research program, we have also found signatures of rapid evolution in a subset of populations studied here. First, in fitness assays performed in control temperature on a subset of populations evolving in this treatment, as well as on their ancestors, we obtained results suggesting that further adaptation to laboratory conditions has proceeded, in populations of both reproductive systems [28]. Furthermore, assaying fertilization trajectories in obligatorily outcrossing populations evolving in both control and elevated temperature, we identified five candidate populations which appear to have evolved increased fertilization rates, relative to their ancestors [29].

Here, we studied population-level fitness under the elevated temperature, assaying populations representing both reproductive systems and both (control and elevated) temperature treatments during experimental evolution, alongside their ancestors. Because the assays were performed in elevated temperature, we expected to find upward fitness divergence in populations evolving in this treatment. Furthermore, we predicted that outcrossing populations should be more prone to adapt–and therefore to show upwards fitness divergence in the assays–due to the benefits of genetic shuffling [3] as well as to the fact that in *C. elegans*, increased temperature poses greater challenge–and thus, may impose a stronger selective pressure—to obligatorily outcrossing than to wild type worms [30]. Importantly, in order to assess the reproducibility of fitness scores, the assays were replicated in 2–4 independent blocks for each population.

## Materials and methods

### 1. Construction of experimental populations

As explained above, we aimed to study the impact of reproductive system on adaptation and diversification of initially (nearly) identical populations. Thus, we started with creating isogenic lines of *C. elegans* strain N2, by 20 generations of single hermaphrodite transfers. Subsequently, we introgressed the *fog-2* gene mutation *q71* independently into each isoline to obtain populations reproducing by obligatory outcrossing (9 cycles of introgression were performed, for details see [30]. The mutation blocks sperm production in hermaphrodites, transforming them into functional females, while male spermatogenesis remains unaffected. This way, for each isoline we obtained a pair of populations differing in reproductive mode (i) androdioecious wild type (henceforth referred to as wt) and (ii) dioecious (henceforth referred to as fog). Apart from the *fog-2* sequence, both populations within each isoline had nearly identical genetic backgrounds.

Three such pairs (referred to as isolines 6, 8 and 9) were used as ancestral populations for experimental evolution (cf. Fig 1, first 2 columns). Each ancestral population was allowed to expand before being split into multiple sub-samples, some of which were banked at -80°C, while the others were divided into two environmental treatments used for the experimental evolution.

Another one of the wt isogenic populations served as a basis for deriving a standard competitor population used in fitness assays (cf. Methods, 3. Fitness assays). Green fluorescent protein (GFP) allele was introgressed into the isoline from strain PD4792, expressing GFP in

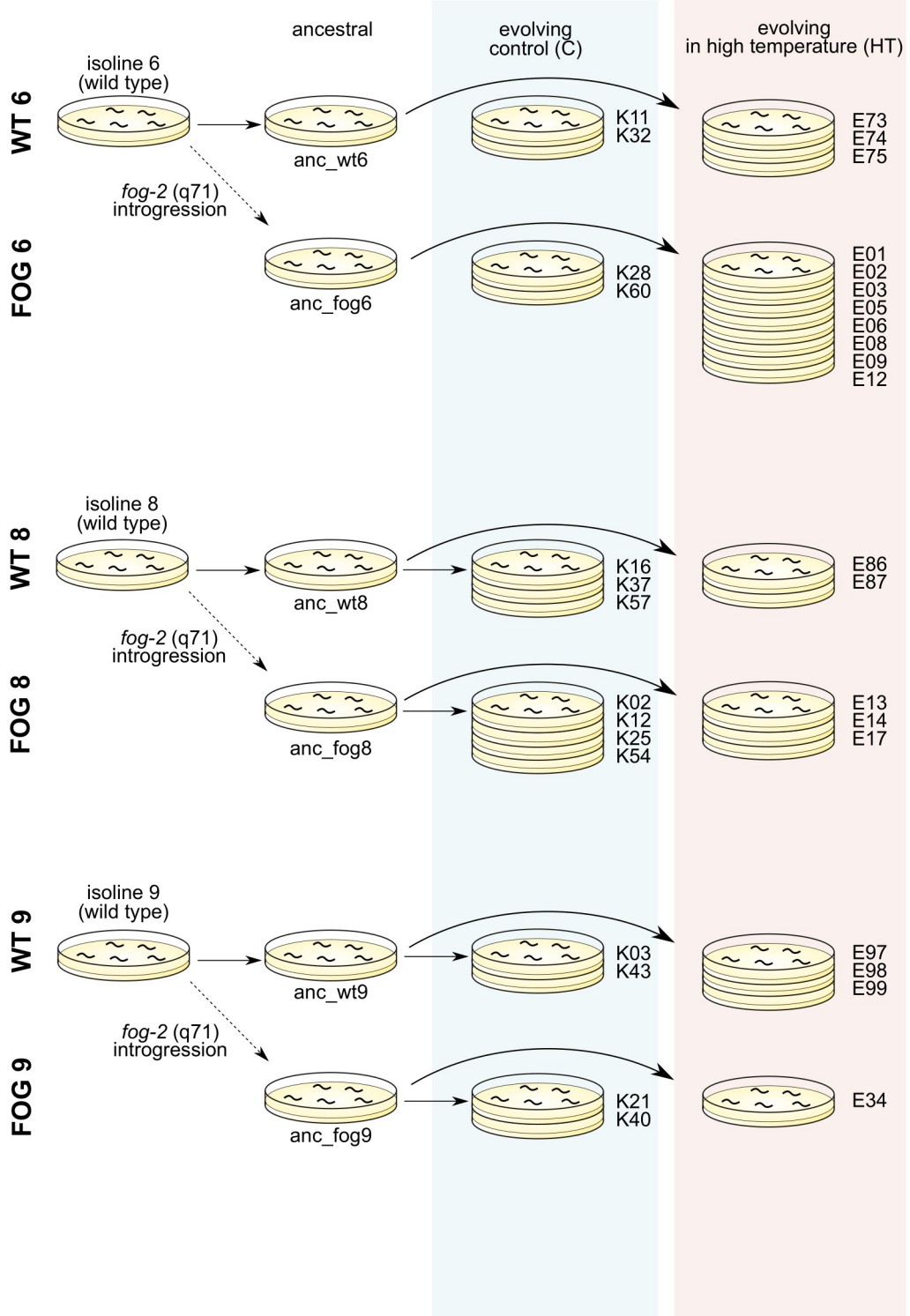

**Fig 1. Creation of populations used in experimental evolution.**

pharyngeal muscles and gut, which was obtained from CGC. The introgression involved 8 cycles of backcrossing, after which the GFP-marked isoline was allowed to expand and subsequently, multiple samples were banked at -80°C.

## 2. Experimental evolution (EE) procedures

We applied two environmental treatments: 20˚C (control, 20˚C being a standard laboratory temperature for *C. elegans* maintenance) and 24˚C (which presents thermal stress to *C. elegans*). Except for the temperature difference, the same standard laboratory conditions were maintained in both treatments.

All EE populations were cultured on 14 cm ø Petri dishes filled with Nematode Growth Medium (NGM) covered with *Escherichia coli* OP50 strain as a food source, and were transferred onto fresh plates every generation, with population size kept at *ca*. 10 000 individuals. Transfers were performed using filters with 15 µm eyelets, which only let early larvae through (L1-L2 developmental stage), retaining older worms as well as eggs; this way, a population can be synchronized. During a transfer, a population plate is washed with 4 ml of S Basal solution [31] and the liquid with suspended worms is transferred to a filter positioned on a test tube. The filtered suspension of L1-L2 larvae is vortexed to achieve uniform distribution of larvae in the liquid and the number of transferred animals is counted in 2–3 drops of 1 l each. Based on this count, the volume of liquid required to transfer 10 000 individuals is estimated and transferred to a fresh plate with bacteria. Transfers were made every *ca*. 3 days in populations kept in 24˚C and every *ca*. four days in populations kept in 20˚C, which referred to one generation cycle. Occasionally, when population state (too few larvae, a fraction of which would additionally be lost during filtering procedures) or compromised team capacity made it impossible to use filtering (which is a highly time-consuming method), chunking method was used as a "back-up" solution for preventing populations' loss. Chunking is a method in which part of the population is transferred onto a new plate in agar chunks. To do this, piece of agar (containing animals) is cut with sterile equipment from the original plate and transferred to a new one [32].

Every *ca*. 12 generations, samples of each population were frozen to enable further assays of phenotypes from different generations: thanks to the ability of the nematodes to survive freezing, a population can be thawed and propagated further even after long-term freezing. This procedure also prevented the loss of EE populations which would otherwise be lost due to cross-contamination, reversal of outcrossing populations to selfing driven by gene conversion [33, 34], or chance events. In such cases, a population was re-started from samples banked at an earlier time point (cf. [34]).

We started with 200 replicate populations (100 from each reproductive type) in the 24˚C treatment and 40 (20 from each reproductive type) in the 20˚C treatment, evenly distributed among the three isolines; we were planning to let them all evolve for at least 200 generations and track their changes in fitness. However, these ambitious plans were met with unrelenting logistic problems, related mostly to cross-contamination among the evolving populations [34], as well as to difficulties involved in maintaining 24˚C populations, especially the obligatorily outcrossing ones (cf. [30] on differences in temperature sensitivity between reproductive types). These problems have first forced us to substantially scale down the experiment, because transfers had to be performed with extra precautions which extended their duration, making it untenable to maintain 240 evolving populations. Subsequently, we detected cross-contamination in some of the populations despite all the precautions, as well as cases of reproductive system reversal by gene conversion [34]. In these cases, we had to use the stock of population samples banked every *ca*. 12 generations to track down the stage before cross-contamination and restart population's evolution from this stage. Ultimately, 41 populations altogether had completed *ca*. 100 or more generations of evolution, 35 of which have been successfully used in fitness assays described below (Table 1 and Fig 1).

## 3. Fitness assays

Fitness assays for all populations were carried out at 24˚C, i.e., the temperature in which the 24˚C populations had been evolving. The assays were carried out in 8 blocks grouped into 4 pairs. Each block included 8–12 EE populations from a single isoline but both environmental treatments and both reproductive systems, along with both ancestral populations (wt and fog) from the appropriate isoline. Each particular combination of populations was assayed twice– hence the pairs. Thus, each EE population was assayed (alongside its ancestral population) in at least 2 independent blocks, with 5 populations assayed in 4 blocks (Table 1). The entire

**Table 1. Experimental Evolution (EE) populations used in the experiment[a].**

| Type | Treatment | Isoline | Population | Block | Generation |
|---|---|---|---|---|---|
| FOG | C | 6 | K28 | b1, b2 | 165 |
| | | | K60 | b1, b2, b7, b8 | 166 |
| | | 8 | K25 | b3, b4 | 165 |
| | | | K12 | b3, b4 | 143 |
| | | | K02 | b3, b4 | 165 |
| | | | K54 | b3, b4 | 164 |
| | | 9 | K21 | b5, b6 | 165 |
| | | | K40 | b5, b6 | 96 |
| | HT | 6 | E01 | b7, b8 | 115 |
| | | | E02 | b1, b2 | 106 |
| | | | E03 | b7, b8 | 114 |
| | | | E05 | b1, b2 | 145 |
| | | | E06 | b1, b2 | 145 |
| | | | E08 | b1, b2 | 145 |
| | | | E09 | b1, b2 | 145 |
| | | | E12 | b7, b8 | 143 |
| | | 8 | E17 | b3, b4 | 144 |
| | | | E13 | b3, b4 | 110 |
| | | | E14 | b3, b4 | 116 |
| | | 9 | E34 | b5, b6 | 112 |
| WT | C | 6 | K11 | b1, b2, b7, b8 | 106 |
| | | | K32 | b1, b2, b7, b8 | 106 |
| | | 8 | K16 | b3, b4 | 106 |
| | | | K37 | b3, b4 | 106 |
| | | | K57 | b3, b4 | 106 |
| | | 9 | K03 | b5, b6 | 106 |
| | | | K43 | b5, b6 | 106 |
| | HT | 6 | E73 | b1, b2, b7, b8 | 146 |
| | | | E74 | b1, b2, b7, b8 | 112 |
| | | | E75 | b1, b2 | 112 |
| | | 8 | E86 | b3, b4 | 112 |
| | | | E87 | b3, b4 | 112 |
| | | 9 | E97 | b5, b6 | 112 |
| | | | E98 | b5, b6 | 112 |
| | | | E99 | b5, b6 | 112 |

[a]Alongside the EE populations, we assayed the wt and fog ancestral populations from isoline 6 (in blocks b1, b2, b7 and b8), isoline 8 (blocks b3 and b4), and isoline 9 (blocks b5 and b6).

assay workflow (below) was identical in each block. This design allowed us to gain insight into the repeatability of our population-level fitness estimates.

We could not assay evolved vs. ancestral populations in direct competition with each other, as they are morphologically indistinguishable. Thus, a GFP-marked wt isoline was used as a standard competitor (see Methods, 1. 1. Construction of starting experimental populations, for details on the population's origins). Fitness of each focal population (evolved as well as ancestral) was assayed in competition with the GFP nematodes (Figs 2 and 3), and the competitive fitness scores of EE populations were subsequently compared with those of their ancestral populations, to calculate divergence scores–measuring, for each EE population, its divergence (difference) in competitive fitness from that of its ancestral population assayed in the same experiment. As the mutation causing GFP expression is dominant, offspring origin in the mixed (focal and GFP-marked) competitive groups can be determined visually: pure focal offspring (whether fog or wt) do not express fluorescence, GFP-GFP as well as mixed GFP-focal offspring express pharyngeal fluorescence.

At the beginning of each block, samples of all participating focal populations, as well as of the competitor GFP population, were thawed from the storage at -80˚C and allowed 2 generations to recover. Subsequently, 4 replicate competition groups were created for each focal (EE or ancestral) population, by mixing L1-L2 larvae from (i) the focal population and (ii) competitor (GFP), at an estimated proportion of 3:1.

To do this, the populations with animals at L1-L2 stage were washed from plates and mixed in falcon tubes (7500 focal animals + 2500 GFP animals), from which a total of 2000 larvae (1500 focal + 500 GFP) were seeded on each of four 6 cm ø Petri dishes (4 replicate competition groups). The numbers of larvae for mixing and seeding were estimated based on counting in 1 l droplets, in the same way as during filter transfers (see above). Thus, the proportion of seeded GFP to non-GFP animals in each competition group was estimated to be 0.75, but because of the possible variance and estimation error, it was checked once again, by taking a sample of the mix from the falcon tube (Fig 2). The sample was mounted on a glass slide and covered with a cover slip. Approximately 10 non-overlapping pictures were done to estimate the initial proportion of focal vs GFP populations and analysed using the automated method based on machine learning taught to distinguish GFP from non-GFP animals [35].

Meanwhile, the competition groups seeded on Petri dishes were allowed 4 days for development and reproduction. After the 4 days, they were washed from the plates with 1 ml S-basal solution and the larvae were separated from the parental generation using the filter method. The filtered liquid (with larvae only) was placed in Eppendorf tubes, from which, after sedimentation, a 5 l drop was taken and placed on a glass slide with a cover slip (Fig 3). Again, 10 non-overlapping pictures were taken and were analysed using the automated method [35]. The proportion of focal (non-fluorescent) larvae in each sample was calculated.

## 4. Data analysis

Fitness scores (W) were first estimated separately for each of the 4 mixed replicate within each focal (EE or ancestral) population, as the natural logarithm of the ratio of two proportions: the proportion of focal individuals in the offspring generation (p1) to the proportion of focal individuals in the parental generation (p0):

$$W = \ln\frac{p_1}{p_0}$$

This score estimates the increase (or decrease) in the proportion of focal individuals over the course of one generation, in competition with the GFP nematodes. In order to

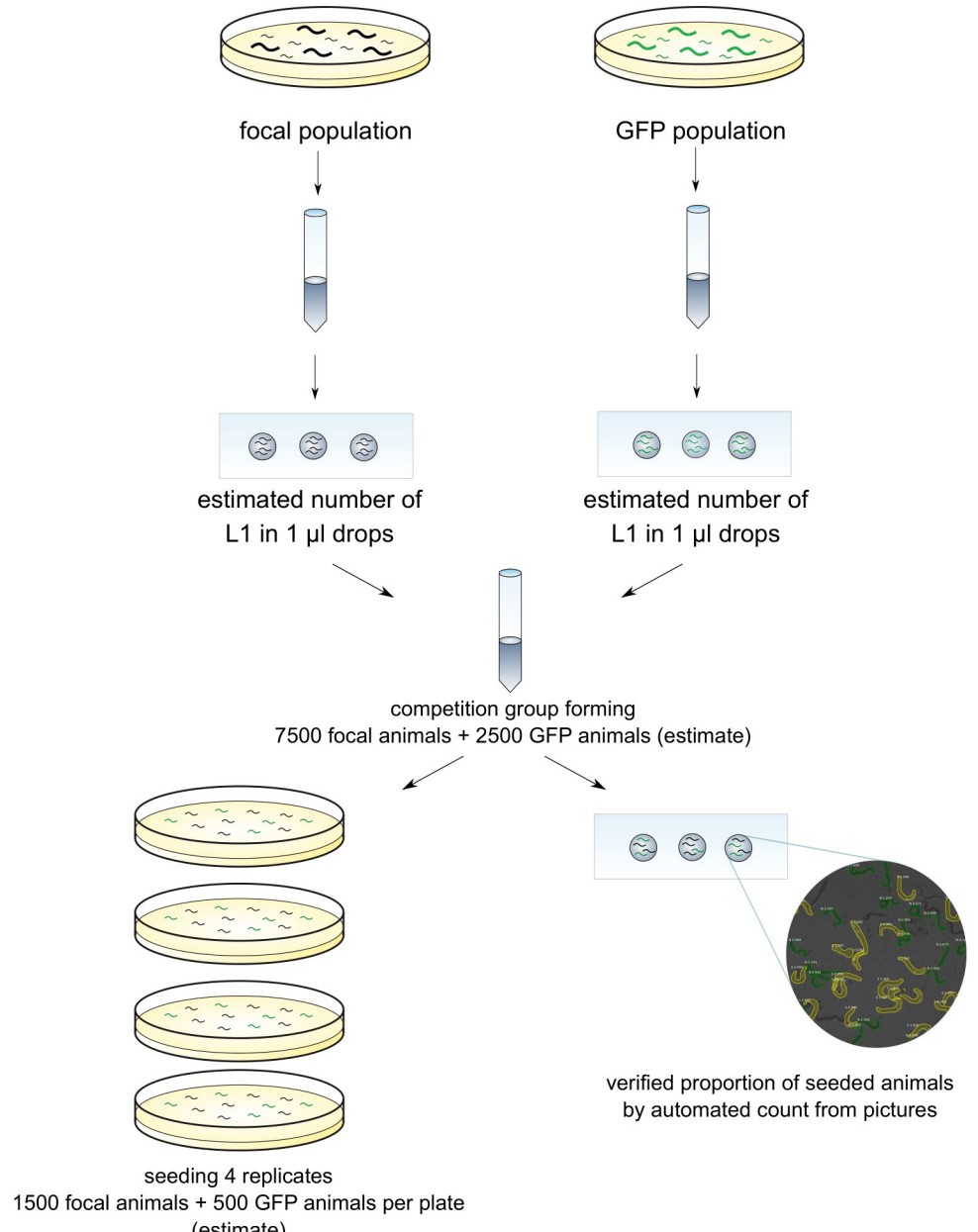

**Fig 2. Preparation of populations seeded for fitness assays.** 'Focal population' denotes one of our experimental populations, either EE or ancestral.

descriptively evaluate the within- and among-block variability of the fitness scores for all populations, we plotted them against population ID, color-coded by replicate block (Fig 4).

Furthermore, for each evolving population we calculated effect sizes assessing its fitness divergence from ancestors, separately for each of the 2 or 4 blocks it was assayed in (cf. Table 1). Calculations were done in such a way that subsequent effect sizes included propagated sampling errors of the constituent values. For each EE and ancestral population within each block it was assayed in, average proportions of focal individuals were calculated separately for parental and offspring generations, across the 4 mixed replicates, and mean fitness score (W) was calculated

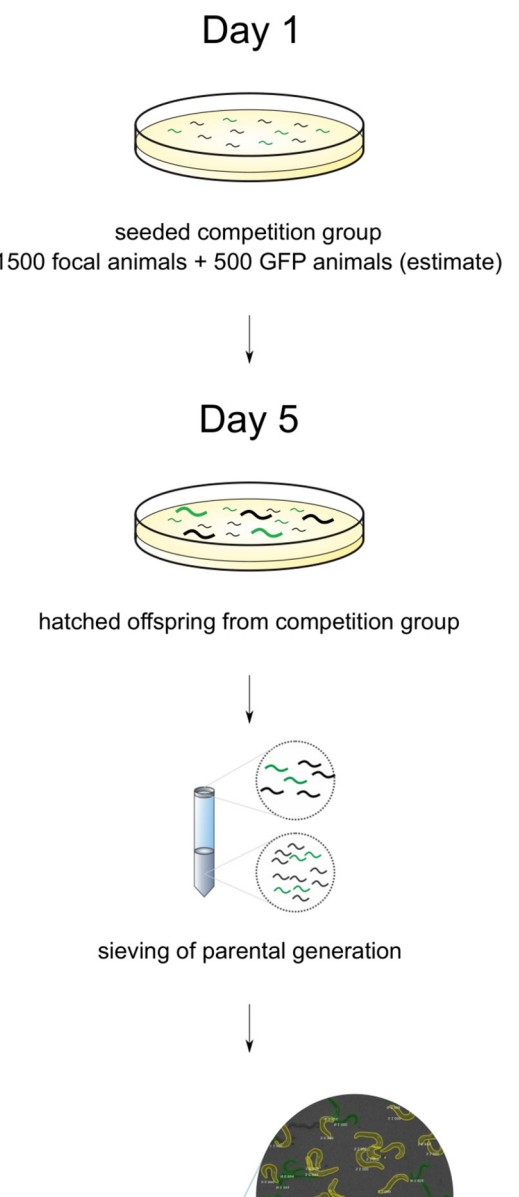

## Day 1

seeded competition group
1500 focal animals + 500 GFP animals (estimate)

## Day 5

hatched offspring from competition group

sieving of parental generation

counting the proportion of focal animals
using automated picture analysis

**Fig 3. Estimation of offspring proportions in fitness assays.**

as the natural logarithm of the offspring-to-parental means ratio. For calculating sampling variances of these means, raw proportions of focal individuals were assumed to be associated with a binomial sampling variance (see S1 Appendix). Sampling variances were calculated using the algebra of variances (i.e., as sums of respective constituent variances, corrected for the inherent

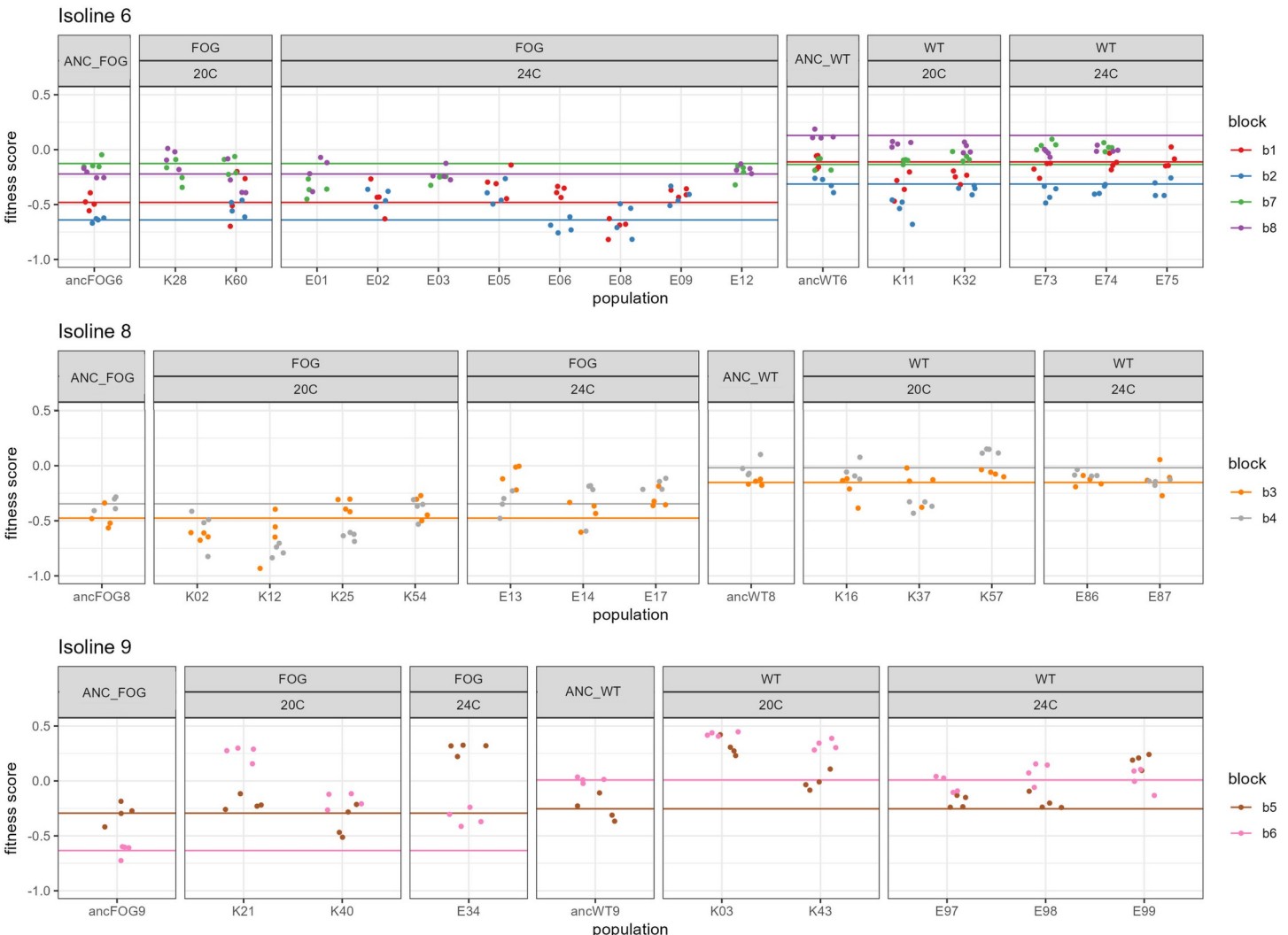

**Fig 4. Raw fitness scores (y axes) for all populations (x axes: Population IDs), color-coded by block ID (legend).** Isoline numbers are shown above the plots; population reproductive systems are shown in top title panels (ANC_ prefix denotes ancestral populations); EE populations' temperature treatments are shown in lower title panels. Horizontal lines show mean values for relevant ancestral populations in each block.

correlation of proportions generated within each block, S1 Appendix). Correction applied was simplified to aid calculations, but it assumed a reasonable average sampling variance of proportion of 0.0005 and a within-block correlation among the 4 mixed replicates = 0.8 (see S1 Appendix for calculation details). Finally, the metric describing divergence of each evolved population from its ancestral state (as assayed in a given block) was calculated as Cohen's d:

$$d = \frac{W_{ee} - W_{anc}}{s_{pooled}} J$$

Where $W_{ee}$ and $W_{anc}$ are mean fitness scores for the evolved and ancestral populations, respectively (calculated for a given block), s_pooled is the pooled standard deviation calculated from the mean fitness scores' sampling variances, and J is a correction factor (see S1 Appendix).

Resulting effect sizes were analysed using a mixed model accounting for the sampling variance of each Cohen's d estimate. The model had population and block as random effects,

isoline, temperature and reproduction mode as fixed factors (including their first and second order interactions), and generation number as a covariate, to control for the fact that populations differed in the number of generations of experimental evolution they had completed. Analysis was performed using the metafor package in R.

All analyses and plots were done in R [36]. The files containing code used for data handling and analyses (including complementary analyses which provided results qualitatively analogous to those described in the paper), along with data files, are accessible from github: https://github.com/szymekdr/230206_Celeg_ee

## Results

Raw fitness data for all populations and blocks are plotted on Fig 4. They show tight clustering within blocks, contrasting with substantial variability among blocks, in the majority of populations, including all 7 those which were assayed in 4 blocks (5 evolved and 2 ancestral, from isoline 6), as well as the remaining 4 ancestral populations from isolines 8 and 9.

Distributions of effect sizes are presented on Fig 5. Model results are presented in Table 2. The model's predicted values, with confidence intervals, from the meta-analytic model are presented on Fig 6. The number of EE generations completed, included as a covariate in the model, showed no significant effect on populations' divergence from ancestors (Table 2); this lack of influence was additionally confirmed by regressing effect sizes on generation number (Fig 7).

Model results (Figs 5 and 6 and Table 2) show that the mean divergence of EE populations from their ancestors differs among isolines, with populations' effect sizes distributed around 0 in isolines 6 and 8, whereas in isoline 9 they were shifted towards positive values (Fig 5, see also Fig 4 for raw fitness scores of evolved vs. ancestral populations). Furthermore, the effects of reproductive system and temperature treatments were also mediated by isoline (Fig 6 and Table 2), albeit the evidence for the latter effect is less robust due to imbalanced distribution of populations among isolines and treatments (Table 1). In isoline 6, overall, the effect sizes were symmetrically distributed around 0 (Fig 5, left panel), with 20˚C wt populations on average slightly tending towards negative fitness divergence from ancestors while no divergence was shown in the remaining three treatment groups (Fig 6, left panel). In isoline 8, overall, the effect sizes were also distributed around 0 (Fig 5, middle panel), with no divergence observed in wt populations; however, in fog 24˚C populations the effect sizes tended towards positive values (suggesting upwards fitness divergence from ancestors) whereas in fog 20˚C populations the opposite tendency was revealed (Fig 6, middle panel). Finally, in isoline 9, the effect sizes were overall shifted to positive values (Fig 5, right panel); surprisingly, this trend was the weakest in wt 24˚C treatment group, with confidence interval overlapping 0 (in contrast to the other three groups; Fig 6, right panel).

## Discussion

We had predicted that populations evolving in the 24˚C treatment (but not those evolving in the control treatment) would be adapting to these conditions, which would manifest as upward fitness divergence from their ancestral populations, when scored in our assays performed at 24˚C. We further predicted that the extent of this divergence may differ between reproductive types, with fog populations being more prone to evolve higher fitness than wt ones. However, our data did not support these predictions.

The first prominent feature of our results was very strong effect of replicate block on populations' fitness scores. As displayed on Fig 4, in most populations, evolved as well as ancestral, the raw fitness scores were closely clustered within replicate blocks while varying considerably among them (Fig 4). This pattern could have been related to various micro-environmental

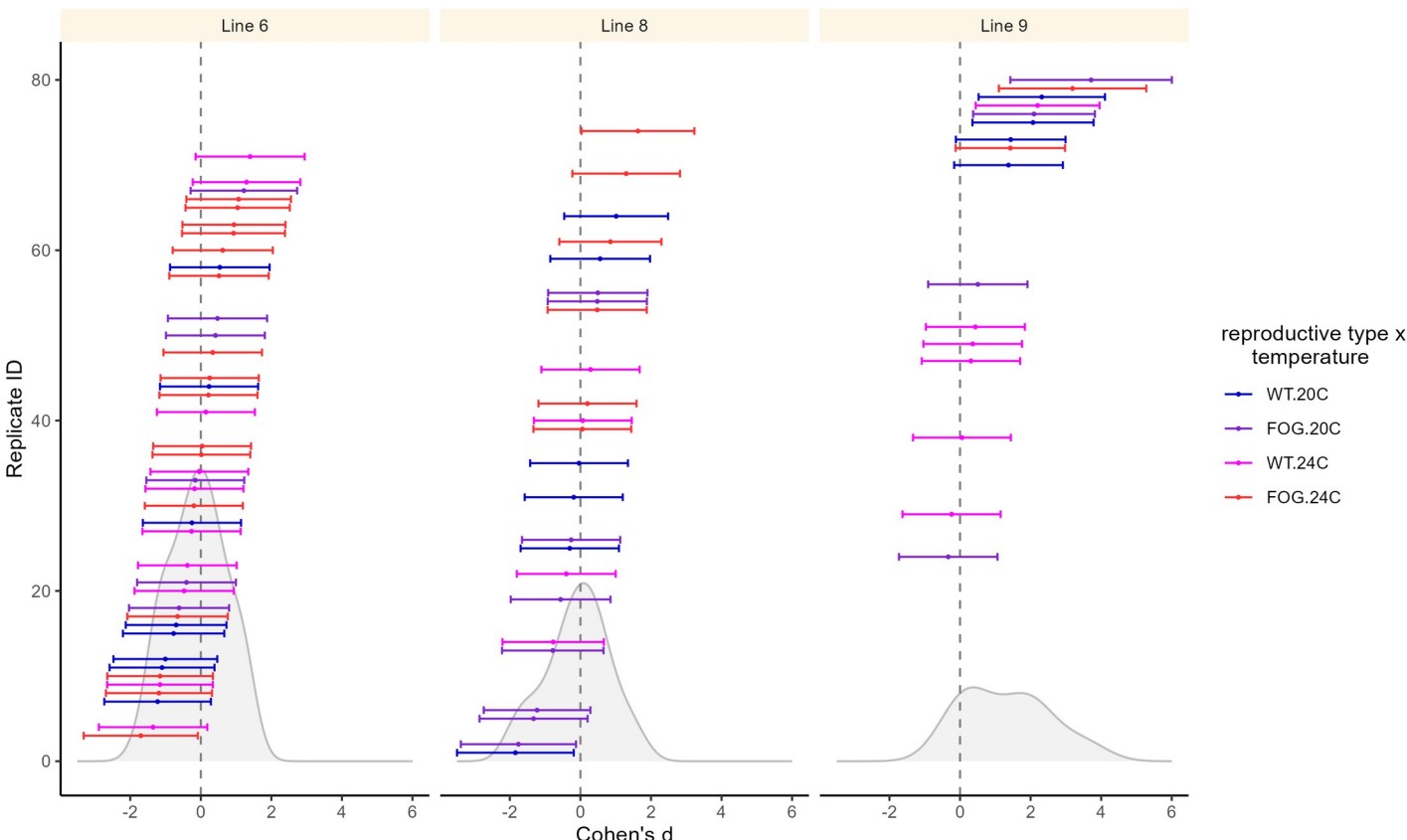

**Fig 5. Effect sizes with confidence intervals based on their sampling variances (calculated as described in Methods & S1 Appendix), separated by isoline and color-coded by reproductive system × EE temperature treatment group.** Frequency distributions of the effect sizes are shown in grey.

**Table 2. Results of the mixed model[a] analyzing the effects of genetic background (isolines 6 *vs.* 8 *vs.* 9), temperature treatment during EE (control: 20˚C *vs.* elevated: 24˚C) and reproductive system (wt *vs.* fog), controlling for the number of generations evolved (generation nr) on EE populations' divergence from ancestors in terms of fitness measured at 24˚C.**

| | estimate | SE | Z value | P value | CI |
|---|---|---|---|---|---|
| Intercept[b] | -1.343 | 0.755 | -1.777 | 0.076 | -2.823 ; 0.138 |
| isoline 8 | 0.433 | 0.411 | 1.054 | 0.292 | -0.3723 ; 1.2381 |
| isoline 9 | 2.255 | 0.505 | 4.462 | 0.000 | 1.2644 ; 3.2452 |
| temperature 24 | 0.238 | 0.378 | 0.630 | 0.529 | -0.5021 ; 0.9777 |
| repr.type fog | 0.171 | 0.565 | 0.302 | 0.763 | -0.9369 ; 1.2783 |
| generation nr | 0.008 | 0.007 | 1.184 | 0.237 | -0.0052 ; 0.0209 |
| isoline 8 × temperature 24 | -0.408 | 0.601 | -0.679 | 0.497 | -1.5859 ; 0.7699 |
| isoline 9 × temperature 24 | -1.623 | 0.639 | -2.542 | 0.011 | -2.8745 ; -0.3714 |
| isoline 8 × repr.type fog | -1.074 | 0.573 | -1.873 | 0.061 | -2.197 ; 0.05 |
| isoline 9 × repr.type fog | -1.104 | 0.763 | -1.446 | 0.148 | -2.5992 ; 0.392 |
| temperature 24 × repr.type fog | -0.018 | 0.612 | -0.029 | 0.977 | -1.2173 ; 1.1817 |
| isoline 8 × temperature 24 × repr.type fog | 1.727 | 0.807 | 2.140 | 0.032 | 0.1452 ; 3.3081 |
| isoline 9 × temperature 24 × repr.type fog | 2.598 | 1.077 | 2.412 | 0.016 | 0.4866 ; 4.7087 |

[a]Variance components associated with random effects in the model were estimated as 0.0000 for population (35 levels), 0.0066 for block (8 levels) and 0.0208 for residuals (80 levels).

[b]The intercept represents wt populations from isoline 6, evolving at 20˚C.

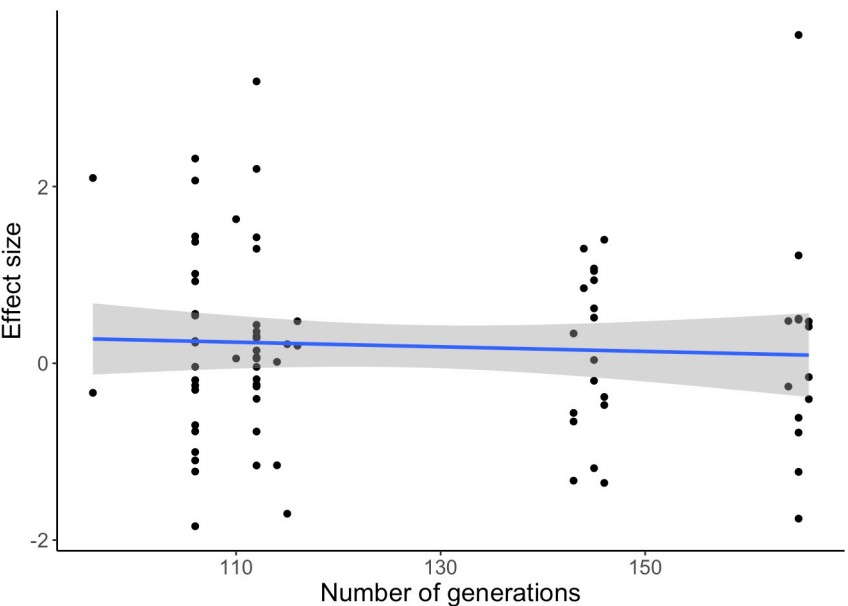

**Fig 6. Predicted values, with confidence intervals, from the mixed model assessing the effects of reproductive system, temperature treatment, ancestral isoline, and their interactions (with population and block included as random variables), on evolving populations' divergence from ancestors.**

**Fig 7. Effect sizes (measuring EE populations' divergence from ancestors) plotted against the number of EE generations completed by the populations.**

factors operating during thawing and acclimation of populations before the assays, possibly compounded by heritable epigenetic effects. For example, initial micro-environmental differences between population samples at the stage of thawing are likely to arise due to high variability in the number of nematodes surviving the freezing, as well as in the presence of bacterial or fungal contaminations. Subsequent transfers preceding the fitness assay are aimed to remove the variation arising from any such effects, by clearing contaminations and calibrating worm density. However, epigenetic responses of *C. elegans* to various environmental stressors can last for many generations after initial exposure (e.g. [37–39]), meaning that environmental variation present at the stage of thawing could still be affecting the nematodes' reproductive output in the assay, even if it has been successfully removed by the standardization procedures in between. Importantly, all abovementioned effects can apply to the competitor as well as the focal populations, further compounding the potential for variability of the competitive fitness scores. Future studies including competitive fitness assays could benefit from using more than one competitor strain, which would allow to parse out the focal vs. competitor inputs into the among-block variability.

The second major finding of our study was that the pattern of the EE populations' divergence from ancestors was considerably different depending on the ancestral isoline (Figs 5 and 6 and Table 2). Notably, it suggests that if only one of these isolines were used in our study, we may have reached one of three entirely different conclusions, depending on which specific isoline it would be. If we had only used isoline 6, we would have seen no average change in population fitness, measured at 24˚C, over the course of experimental evolution, regardless of the reproductive system × temperature treatment group. This could have led us to conclude that within the time-frame of our experimental evolution there was probably too little supply of relevant *de novo* mutations to enable response to selection (since our experimental evolution started with isogenic ancestral population, it had to rely on *de novo* mutations). On the other hand, if we had studied isoline 8, we would have concluded that fog, but not wt populations evolving at 24˚C showed increased fitness in this temperature compared with their ancestors, supporting the prediction that outcrossing facilitates adaptation. We would have further observed that fog, but not wt populations evolving at 20˚C showed fitness decline, relative to their ancestors, when assayed in the higher temperature. Finally, if we had only studied isoline 9, we would see positive fitness divergence in evolved populations regardless of the reproductive × temperature treatment group, which could be open to speculations, but suggests that there might have been adaptation to some other, uncontrolled, environmental factors (cf. [28]). These results suggest that the populations' evolutionary responses have been profoundly affected by differences in the ancestral genetic background. Since the ancestral isolines had been derived from the N2 strain obtained from CGC, which is genetically highly homogeneous, these differences would have to arise primarily or exclusively from *de novo* mutations accumulated during the process of deriving the ancestral populations.

Alternatively, however, the observed differences among isolines might be an artifact produced by the high levels of unexplained among-block variation discussed above. In our design, blocks were nested within isolines (Fig 4 and Table 1). In the meta-analytic model, block was included as a random effect, thus, the significance of the isoline effects was estimated taking into account the variation associated with blocks. However, given the small number of blocks, these estimates need to be treated with caution.

Whether the differences we found among isolines were indeed due to genetic background or an artifact of the unidentified environmental factors driving high among-block variation, our data strongly support the calls for paying much more regard to biological variability, of various sources, when designing studies and formulating research claims (e.g. [40, 41]). As Voelkl et al. [41] point out in their Perspective, while abundant biological variation belongs to the most

fundamental characteristics of life, the gold standard of current laboratory practices is stripping this variation away as much as possible, through rigorous standardization of both organisms and their environment. Voelkl and colleagues argue, and we have come to agree, that such approach is in fact based on incorrect assumptions, which they call "the myth of a pure treatment effect that 'emerges' as more sources of variation are eliminated." The myth is based on fundamental misunderstanding of the complex nature of biological phenomena, where the effects of any particular genetic or environmental variable are almost always dependent on the interplay of a plethora of other variables. Thus, when variability other than the treatment(s) of interest is minimized in experimental studies through standardization practices, instead of the "pure" effect of the treatment(s) we obtain a highly idiosyncratic one, contingent on the very particular conditions of our particular study. Furthermore, even very strict standardization protocols usually fail to eliminate all sources of external variation such as, e.g., differences in quality of reagents, fluctuations in a microenvironment or idiosyncratic techniques of researchers [42]. In the study reported by Lithgow et al. [42], scientists from three different laboratories took measures to decrease variability between the labs and performed the same experiment concerning lifespan of *Caenorhabidits* nematodes. It turned out that even after establishing identical protocols, and identical laboratory equipment, the variation between laboratories, albeit decreased, was still present and affecting the lifespan of the animals [42]. In our study, the high among-block variability of fitness scores, discussed above, was found despite great efforts to standardize the conditions of fitness assays across all replicate blocks.

## Conclusions

*"We often study individual species or communities (. . .); results can be strongly context dependent, full of idiosyncrasies. They may be generalized, but only to a limited extent (. . .) carrying out an analogous study on another species, or on the same species in a different place or year, may yield different results."* [40]

*"Since variation is a fundamental property of any population of organisms, treatment effects can be assessed and interpreted meaningfully only against biological variation—including gene × environment interactions. Owing to context- dependent variability in responses to treatment (. . .), there is no such thing as a pure treatment effect for a population of living organisms. (. . .). Studies that are too narrowly defined cannot reliably be generalized: if only males are included, the results may differ in meaningful ways in females; the responses of a single inbred strain may not hold for other strains"* [41]

In consequence of the issues outlined above, strict standardization practices in experimental studies [41], along with the common practice of over-generalizing results obtained for specific study populations and conditions [40], are probably among important contributors to the problem of poor reproducibility of research results, currently plaguing many scientific disciplines (cf. [43–47]). In the Introduction, we discussed how insights concerning the dynamics of adaptive evolution, derived from studies on microbes, should not be easily extrapolated on multicellular organisms [7]. Here we would like to update this claim by cautioning against any kind of generalizations based on study outcomes obtained for a specific, narrow set of genetic and environmental conditions. In experimental studies, Voelkl and colleagues propose that instead of standardization, the opposite approach should be implemented in order to improve the validity of research findings: deliberate heterogenization of genetic and environmental conditions. Such approach enables assessing both the robustness (consistency) of treatment outcomes against the variable background, and, conversely, their dependency on other

conditions. The obvious problem is that including a wide range of genetic and environmental variation as a background for assessing the treatment(s) of interest in any particular experiment may be logistically unmanageable. The challenge, therefore, is how to find the right balance between the complexity of biological phenomena we are trying to understand and logistical feasibility of studying them.

## Supporting information

**S1 Appendix.**
(DOCX)

## Acknowledgments

We thank Wiesław Babik and other members of Genomics and Experimental Evolution Group for moral support, advice and comments on earlier versions of the manuscript.

## Author Contributions

**Conceptualization:** Marta K. Labocha, Zofia M. Prokop.

**Data curation:** Joanna K. Baran, Paulina Kosztyła.

**Formal analysis:** Szymon M. Drobniak.

**Funding acquisition:** Marta K. Labocha, Zofia M. Prokop.

**Investigation:** Joanna K. Baran, Paulina Kosztyła, Weronika Antoł, Marta K. Labocha, Karolina Sychta, Zofia M. Prokop.

**Methodology:** Joanna K. Baran, Paulina Kosztyła, Weronika Antoł, Marta K. Labocha, Karolina Sychta, Zofia M. Prokop.

**Project administration:** Karolina Sychta.

**Software:** Joanna K. Baran.

**Supervision:** Zofia M. Prokop.

**Validation:** Joanna K. Baran, Zofia M. Prokop.

**Visualization:** Joanna K. Baran, Weronika Antoł, Szymon M. Drobniak.

**Writing – original draft:** Joanna K. Baran, Szymon M. Drobniak, Zofia M. Prokop.

**Writing – review & editing:** Joanna K. Baran, Paulina Kosztyła, Weronika Antoł, Marta K. Labocha, Karolina Sychta, Szymon M. Drobniak, Zofia M. Prokop.

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
