## [Decision Letter · Decision Letter 0]

6 Dec 2023

PONE-D-23-36678Genetic background modulates the effects of reproductive system and temperature in experimentally evolving Caenorhabditis elegans populationsPLOS ONE

Dear Dr. Prokop,

Thank you for submitting your manuscript to PLOS ONE. After careful consideration, we feel that it has merit but does not fully meet PLOS ONE’s publication criteria as it currently stands. Therefore, we invite you to submit a revised version of the manuscript that addresses the points raised during the review process.

We look forward to receiving your revised manuscript.

Kind regards,

Ilya Ruvinsky

Academic Editor

PLOS ONE

Journal Requirements:

3. Thank you for stating the following financial disclosure: "National Science Centre, Poland, grants OPUS (2013/09/B/NZ8/03317) and Sonata Bis (UMO-2017/26/E/NZ8/00879) to Z.M.P.".

4. Thank you for stating the following in the Acknowledgments Section of your manuscript: "We thank Wiesław Babik and other members of Genomics and Experimental Evolution Group for moral support, advice and comments on earlier versions of the manuscript. This work was financed by the Polish National Science Centre grants OPUS (2013/09/B/NZ8/03317) and Sonata Bis (UMO-2017/26/E/NZ8/00879) to Z.M.P.".

Please remove any funding-related text from the manuscript and let us know how you would like to update your Funding Statement. Currently, your Funding Statement reads as follows: "National Science Centre, Poland, grants OPUS (2013/09/B/NZ8/03317) and Sonata Bis (UMO-2017/26/E/NZ8/00879) to Z.M.P.".

Additional Editor Comments:

Dear Dr. Prokop,

The two expert reviewers who evaluated your manuscript agreed that your study was interesting and potentially suitable for publication in PLoS ONE.

They also identified several aspects of data analysis, interpretation, and description of results that require modification.

If you wish to have your manuscript to be further considered by PLoS ONE, please address all of their comments thoroughly. In particular, the claims related to the "genetic background" were seen by the reviewers as insufficiently established.

Please accompany your revised manuscript with a detailed list of how each of the reviewers’ criticisms was addressed by you.

Reviewers' comments:

Reviewer's Responses to Questions

**Comments to the Author**

1. Is the manuscript technically sound, and do the data support the conclusions?

Reviewer #1: Partly

Reviewer #2: Yes

2. Has the statistical analysis been performed appropriately and rigorously? 

Reviewer #1: Yes

Reviewer #2: Yes

3. Have the authors made all data underlying the findings in their manuscript fully available?

Reviewer #1: Yes

Reviewer #2: Yes

4. Is the manuscript presented in an intelligible fashion and written in standard English?

Reviewer #1: Yes

Reviewer #2: Yes

5. Review Comments to the Author

Reviewer #1: This is a very worthwhile study, for a number of reasons, and it is important to publish. Everyone contemplating experimental evolution with C. elegans (or anything else, for that matter) should read and take it to heart. Similarly, anyone interested in the experimental quantification of competitive fitness should read and take to heart. PLoS One seems like an appropriate venue.

As much as anything, the authors demonstrate that the Road to Hell is paved with good intentions. They set out to be the Lenskis of the multicellular world, and they found it's not so easy. Of their hoped-for 200 lines evolved for 200 generations, partitioned into starting isolines, temperature and mating systems, after all is said and done they report results from 35 lines evolved for an average of 127 generations, distributed unevenly across treatments.

The first key feature of this work is shown clearly in Figure 4. Looking at the fog-2 ancestor of isoline 8 at 20 deg, the estimates of W cluster tightly within blocks. Similar patterns hold for other ancestors. These lines are (nearly) isogenic, as is the competitor strain**. Thus, the variance in these assays must be due to heritable environmental effects (which are common in C. elegans) in the focal strain, the competitor, or (very likely) both.

** We are not given any details about the competitor strain beyond "...a green fluorescent protein (GFP)-marked strain was used as a standard competitor". What was the strain? How inbred was it? How was the competitor propagated among blocks? Was a large population cryopreserved in many aliquots at the same time and one aliquot thawed for each assay block? Was the competitor cryopreserved from multiple plates, maybe at different times? These are important details, which we are not told.

Which leads to the second key feature, which is that isoline is confounded with block(s). Isoline 9, which is the only one of the three that consistently evolved, was assayed in blocks 5 and 6. If for some reason both the ancestor and the competitor strain was compromised in those two blocks but the evolved lines were not – which is certainly possible – the increase in fitness would be a mirage. The fact that isoline 9 in the fog-2/heat treatment is represented by only a single population (E34) does not help matters any.

And also, there are only four biological replicates per population, and no more than eight populations per treatment, and sometimes as few as two. Four replicates per population is reasonable if the point is to (say) estimate the among-line variance of a large set of inbred lines or wild isolates. It is noteworthy, and interesting, that the among-block variance is consistently greater than the among-population variance within a block, of which the point estimate is 0.0000 (Table 2). Since the unit of evolution in this experiment was the population, taken at face value, that result seems strange. Either (a) all the populations within a group evolved in identical ways – which could happen if the infinitesimal model of quantitative genetics is true – or (b) there is insufficient power to detect a relatively small component of among-population variance. But clearly, whatever the cause of variation among blocks, it is large compared to the variation among populations, and among replicates within populations.

Which brings me to the authors headline claim, that there is an "Effect of genetic background", because one isoline (9) appears to respond to selection, whereas the other two do not. Given that these are "isolines", evolution must be due to new mutations rather than the increase in frequency of segregating variants. In essence, the authors are asking us to believe that isoline 9 is more evolvable than the other two. There are three possible (not mutually exclusive) reasons for differences in evolvability: (1) differences in mutation rate, (2) a different distribution of fitness effects (i.e., epistasis), or (3) random chance. The three isolines were constructed by 20 generations of single-worm transfer from an N2 ancestor. So the genetic variation among isolines will be 20 gens of mutation accumulation, plus 9 generations of introgression with fog-2 (or not), plus whatever segregating variation was present in the N2 progenitor. Then, we are asked to believe that populations derived from one of the isolines evolved – all in the same way, see the preceding paragraph – whereas populations derived from the other two isolines did not. Given that N2 is already highly inbred, and that fog-2 is initially in an N2 background, and that 20 gens of MA will introduce about 6-8 new mutations per-genome, I doubt the three isolines are very different genetically. Which is to say, given the large variation among blocks and the confounding of block with isoline, and the sketchy description of how the competitor was handled, I very much doubt that the difference of isoline 9 has anything at all to do with its genetic background. Random chance sounds way more plausible.

So here's the bottom line. This is an important paper for anyone interested in experimental evolution in Caenorhabditis, and I want to see it published. I have experienced most (if not all) of the frustrating aspects of this project in my own 20 years of working with experimental evolution with Caenos, in particular the perfidy of variability among isogenic lines across fitness assay blocks. It is now abundantly clear that short-term heritable epigenetic effects are both ubiquitous and extremely important in C. elegans, and that the problem is squared with competitive fitness assays because the effects apply equally to the competitor strain. In fact, in reading this paper I reached the conclusion that competitive fitness assays should always include two independent competitor strains as a control, which would allow among-block variability to be attributed either to the focal strain or (one of) the competitors.

signed,

-Charlie Baer

Reviewer #2: Baran et al. report the results of a massive experimental evolution study in C. elegans. Starting with very little standing genetic variation, the presence of adaptation depended primarily on the starting strains (after 96-165 generations in either selfing or outcrossing lines reared at two different temperatures). This work is a valuable contribution to the literature. I can find no obvious problems with this manuscript (although I do not have the expertise to evaluate the Appendix). Below are my minor comments.

Discussion and motivation. I appreciate the references to the Lenski LTEE. I have also thought that an analogous study in nematodes could potentially be a powerful inroad towards understanding how evolution works in animals. This clarifies my confusion regarding a similar paper that was recently published (Palka et al. 2023 PeerJ)-- the aim was to potentially evolve populations indefinitely (like the LTEE). This leads to my major issue with the discussion. It is largely oriented around the need for replication in biological experiments. I agree with much of this language, and I think replication should be considered in the interpretation. Yet, I was surprised by how little language was devoted to how the original isolines are different and what might be next for these EE populations. Previous literature has suggested there exists some cryptic genetic variation in "wild-type N2" lines among laboratories (Gems & Riddle 2000 J. Gerontology Series A; Vergara et al. 2009 BMC Genomics). What is the extent of genomic variation among the ancestral isolines, and what new mutations emerged in isoline 9 that could potentially drive adaptation in this background? And, what are the future plans for these EE populations?

Figures. In Figure 4, it might be useful to have a dotted horizontal line to note the mean ancestral fitness for each line/temperature/reproduction group. As that is the most important comparison, I was struggling to visualize that key difference across the various panels of this figure. Additionally, Figures 4-5 still have default axis ("ln_prop") or legend ("repro.temp") labels that, while recognizable, are disheveled.

Very minor comments

Line 56. Should this be "Not only do both of these features..." ?

Line 200. "Apart from the fog-2 sequence..." Also fog-2 should be italicized.

Line 218-221. Can embryos also pass through these filters?

Line 240-265. How often do fog-2 gene conversion events occur? Also, reference 35 does not have a journal listed.

Line 250. Should this be "met with" instead of "verified by"?

Line 461. Should this be "wild-type" instead of "wt"?

Thanks for sharing this work.

6. PLOS authors have the option to publish the peer review history of their article (what does this mean?). If published, this will include your full peer review and any attached files.

Reviewer #1: No

Reviewer #2: No

---

## [Author Response · Author response to Decision Letter 0]

9 Feb 2024

Accordingly to the directions received in the email, we uploaded the responses as a file labeled 'Response to Reviewers'

---

## [Decision Letter · Decision Letter 1]

26 Feb 2024

Reproductive system, temperature, and genetic background effects in experimentally evolving populations of Caenorhabditis elegans

PONE-D-23-36678R1

Dear Dr. Prokop,

We’re pleased to inform you that your manuscript has been judged scientifically suitable for publication and will be formally accepted for publication once it meets all outstanding technical requirements.

Kind regards,

Ilya Ruvinsky

Academic Editor

PLOS ONE

Additional Editor Comments (optional):

Thank you for addressing reviewer comments as well as you did. I am happy to recommend your work for publication in PLoS ONE as a valuable contribution to knowledge in the C. elegans community. Congratulations.

Reviewers' comments:

Reviewer's Responses to Questions

**Comments to the Author**

1. If the authors have adequately addressed your comments raised in a previous round of review and you feel that this manuscript is now acceptable for publication, you may indicate that here to bypass the “Comments to the Author” section, enter your conflict of interest statement in the “Confidential to Editor” section, and submit your "Accept" recommendation.

Reviewer #1: All comments have been addressed

Reviewer #2: All comments have been addressed

2. Is the manuscript technically sound, and do the data support the conclusions?

Reviewer #1: Yes

Reviewer #2: Yes

3. Has the statistical analysis been performed appropriately and rigorously? 

Reviewer #1: Yes

Reviewer #2: Yes

4. Have the authors made all data underlying the findings in their manuscript fully available?

Reviewer #1: Yes

Reviewer #2: Yes

5. Is the manuscript presented in an intelligible fashion and written in standard English?

Reviewer #1: Yes

Reviewer #2: Yes

6. Review Comments to the Author

Reviewer #1: The authors have done an exceptionally good job of replying to my comments. I especially commend them for being so forthright about cross-contamination of the stocks. I have had to do that on occasion, and it's painful. My only comment in that regard is that I think contamination would have been much less likely to occur if transfers had been done by chunking from plates rather than using the filters. Many fewer moving parts, and you can flame a spatula in a bunsen burner between transfers. If the authors agree with that thought, they might consider adding a cautionary sentence making that point. Or not.

-Charlie Baer

Reviewer #2: In my opinion, the authors have satisfactorily addressed the reviewer comments. I still think this work will be a valuable contribution to the literature. Thanks again for sharing this work.

7. PLOS authors have the option to publish the peer review history of their article (what does this mean?). If published, this will include your full peer review and any attached files.

Reviewer #1: No

Reviewer #2: No

---

## [Editor Report · Acceptance letter]

22 Mar 2024

PONE-D-23-36678R1 

PLOS ONE

Dear Dr. Prokop, 

I'm pleased to inform you that your manuscript has been deemed suitable for publication in PLOS ONE. Congratulations! Your manuscript is now being handed over to our production team.

Kind regards, 

on behalf of

Dr. Ilya Ruvinsky 

Academic Editor

PLOS ONE